# Pharmacological Treatment of Perianal Gland Tumors in Male Dogs

**DOI:** 10.3390/ani13030463

**Published:** 2023-01-28

**Authors:** Adam Brodzki, Wojciech Łopuszyński, Piotr Brodzki, Katarzyna Głodkowska, Bartosz Knap, Paulina Gawin

**Affiliations:** 1Department and Clinic of Animal Surgery, Faculty of Veterinary Medicine, University of Life Sciences in Lublin, 20-612 Lublin, Poland; 2Department of Pathomorphology and Forensic Veterinary Medicine, University of Life Sciences in Lublin, 20-612 Lublin, Poland; 3Department of Andrology and Biotechnology of Reproduction, Faculty of Veterinary Medicine, University of Life Sciences in Lublin, 20-612 Lublin, Poland; 4Doctoral School, Medical University of Lublin, 20-093 Lublin, Poland; 5Chair and Department of Experimental and Clinical Pharmacology, Faculty of Medicine, Medical University of Lublin, 20-090 Lublin, Poland; 6Cedrob SA, Ujazdówek 2A, 06-400 Ciechanów, Poland

**Keywords:** perianal gland tumors, male dogs, treatment, antihormonal therapy

## Abstract

**Simple Summary:**

Perianal gland tumors are among the most common tumors in sexually intact male dogs. They are hormone-dependent tumors expressing androgen receptors (ARs) and estrogen receptors (ERs). Therefore, the treatment of choice for benign perianal tumors is castration. Surgical treatment is indicated in large benign ulcerated tumors, in malignant tumors, or in cases of recurrence. An alternative method may be pharmacological treatment by administering drugs that block hormone receptors. In the conducted studies, complete tumor regression was obtained in benign tumors and partial regression was obtained in malignant tumors after the administration of tamoxifen and cyproterone acetate, which are pharmacological hormone receptor blockers. The obtained results indicate that pharmacological antihormonal therapy may be considered as an alternative or neoadjuvant therapy for male dogs with perianal tumors.

**Abstract:**

The presence of androgen (AR) and estrogen (ER) receptors has been demonstrated both in normal perianal (hepatoid) glands and in perianal tumors. The aim of this study was to demonstrate the relationship between the expression of AR and ER in perianal gland tumors and the effectiveness of antihormonal treatment. The study was performed on 41 male dogs with neoplastic lesions of the anal region. Histopathological evaluation of the lesions revealed 24 adenomas, 12 epitheliomas, and five carcinomas. Treatment was administered orally with tamoxifen at a dose of 1 mg/kg BW and cyproterone acetate at a dose of 5 mg/kg. Tumor diameters were measured regularly with calipers and recorded in millimeters starting with the measurement before treatment, and then after 1, 2, 3, 6, 12, 18, and 24 months of therapy. The results show that hepatoid adenomas that are characterized by high expression of AR and ER receptors respond positively to antihormonal therapy, resulting in complete tumor regression. For locally malignant hepatoid epitheliomas and carcinomas with low expression of AR and ER receptors, antihormonal therapy makes it possible to reduce the size of the tumor, but does not make it possible to cure it completely.

## 1. Introduction 

There are three major glandular anatomical structures located in the anal region of dogs where the neoplastic process can develop; these include the anal glands (*glandulae anales*), the glands of the rectal sinuses (*glandulae sinus paraanalis*), and the perianal (hepatoid) glands (*glandulae circumanales*). The latter are modified sebaceous glands located in the skin around the anus, foreskin, and base of the tail, along the back, and in the groin area and inner side of the thighs. The process of neoplastic transformation most often involves the perianal glands, and the most common form in terms of histopathology is adenoma. Tumors of the anal area in male dogs are a serious health problem; in terms of incidence, they rank third among all types of tumors occurring in males of this animal species. They are second only to skin and testicular tumors [1,2]. According to the literature, they account for 9–18% of all skin cancers reported in male dogs [3,4,5,6,7]. The literature shows that tumors of the perianal region are among the hormone-dependent tumors to both androgens and estrogens. These hormones, after combining with their specific receptors in the glandular tissue, stimulate rapid cell division and induce the process of carcinogenesis. The presence of androgen and estrogen receptors, both in healthy glandular tissue and in tumor-lesioned tissue, has been confirmed in many studies [4,8,9,10]. Among other things, it was shown that the percentage of androgen and estrogen receptors was significantly higher in neoplastic cells than in healthy glandular tissue, indicating that they are involved in the development of the cancer process [4,8,9,10]. Our previous studies confirmed the presence of both estrogen and androgen receptors in all types of tumors of the perianal glands, as well as the different expression of these receptors depending on the type of tumor [11]. Attempts were made to treat the described tumor lesions with the use of antihormonal preparations [12,13]. Various therapeutic methods have been used in the treatment of perianal tumors. Some authors pointed to cryosurgery as the most effective treatment method, arguing that it is easy to perform and most often associated with a low risk of complications in the form of hemorrhage. In addition, the recurrence rate of the tumor process when treated with cryosurgery is low, which gives a good prognosis [14]. An alternative method is the use of a combination of chemotherapeutic injections and an electrical pulse directed into the interior of the tumor, which allows for the effective treatment of tumors of larger sizes but requires specialized hygiene following the procedure due to the extensive necrosis of tumor tissues that develops after the procedure [10,15]. In the case of the hormonal treatment and castration, the effectiveness of this method of treating malignant tumors has not been confirmed, as good results were achieved only when the tumor mass was radically removed with a large margin [10,15]. The use of the aforementioned treatment methods involves the risk of complications and other inconveniences. Most methods require the use of general anesthesia with associated anesthetic risks, which are particularly high for older dogs, who are most often affected by this condition. In addition, in the case of surgical removal of tumors, the recovery period is a discomfort for the patient and the owner; thus, the concept of treatment with antihormonal drugs offers a chance for effective treatment without high risks. This paper describes the effects of treating perianal gland tumors with hormone receptor blockers depending on the type of tumor, as well as on the expression of AR and ER receptors.

## 2. Material and Methods 

The study was performed on 41 male, non-orchiectomized dogs aged 4–17 years. Mixed-breed dogs—16 (67%), Cocker Spaniels—6 (15%), Wire-Haired Fox Terriers—3 (7%), Dachshunds—2 (5 %), German Shepherds—2 (5%), and Miniature Pinscher—2 (5%) were the most common breeds in this study. The remaining breeds were represented by one individual. The animals were patients treated for neoplastic lesions of the anal region at the Department of Animal Surgery of the University of Life Sciences in Lublin. After taking a history and performing a clinical evaluation of the lesions, tumor sections were taken from the diagnosed animals for histopathological examination and immunohistochemical assessment of AR and ER receptors using a 0.6 cm diameter skin trepan. Histopathological examination carried out according to the histological classification of skin tumors according to AFIP/WHO revealed 24 adenomas, 12 epitheliomas, and five carcinomas [16]. The immunohistochemical staining showed mainly nuclear expression of AR and ER in the neoplastic cells. Both the androgen and the estrogen receptors were expressed in adenoma, epithelioma, and carcinoma cases. However, the highest expression of the receptors was revealed in adenoma, and then in epithelioma, representing a low-grade malignant neoplasm. In the case of carcinoma, the expression of sex hormone receptors was very weak. Detailed results of histopathological examination and immunohistochemical determinations of AR and ER receptors in tumors from dogs included in the present study have been published [11]. After histopathological evaluation of the lesions, antihormonal pharmacological therapy was implemented. The animals were divided into three groups depending on the results of histopathological examination, i.e., adenoma, epithelioma, and carcinoma groups. Treatment with the active substance tamoxifen (tamoxifen preparation), at a dose of 1 mg/kg BW orally in tablet form, and cyproterone acetate (Androcur), at a dose of 5 mg/kg BW also orally, was used. During the course of the study, tumor diameters were measured regularly with electronic calipers by the same person and recorded in millimeters starting with the measurement before treatment, and then after 1, 2, 3, 6, 12, 18, and 24 months of therapy.

### Statistical Analysis

Statistical analyses and figure generation were performed using Statistica 13.3 software (StatSoft Polska, Cracow, Poland) and GraphPad Prism 8.3.0 software (GraphPad Software, Boston, MA, USA). The analysis of data distribution showed that the obtained dataset was not normally distributed. Consequently, nonparametric statistical tests were used. All data were expressed as the median with the interquartile range. Changes in tumor diameter during treatment within each experimental group were compared using the Friedman test for multiple comparisons. For clarity of presentation, tumor reductions in consecutive months of treatment were expressed as the percentage change in tumor diameter from the baseline value. Percentage changes in tumor diameter between experimental groups (within each treatment interval) were analyzed using the Kruskal–Wallis test. For all statistical analyses, three levels of probability value (*p*-value) were used: 0.05, 0.01, and 0.001.

## 3. Results 

The results of measuring the diameter of tumors at different stages of the experiment are shown in Table 1. 

The diameter of tumors in dogs in all groups was the largest when measured before the start of treatment. Over the course of the treatment, the diameter of the tumors decreased, illustrated in Figure 1 as the percentage reduction in tumor diameter relative to the initial tumor diameter. 

In the group of dogs diagnosed with adenoma, the tumor diameter was the largest before the start of antihormonal treatment and decreased significantly in the subsequent months of therapy. After 12 months of antihormonal treatment, the tumors in this group had completely disappeared and there was no recurrence over the following year. In the group of dogs with adenoma, differences in tumor diameter during treatment compared to baseline size had statistically significant values for results after 3, 6, 12, 18, and 24 months of the treatment (*p* < 0.001) (Table 1). The results of measuring tumor diameter in the epithelioma group of dogs showed the largest values before the treatment. Tumor diameter was statistically significantly reduced relative to pretreatment tumor diameter after 3, 6, 12, 18, and 24 months of antihormonal treatment (*p* < 0.001). In the epithelioma group of dogs, tumor diameter was significantly reduced over the course of treatment; however, full recovery was not achieved in all dogs in the group (Figure 1). In the carcinoma group of dogs, tumor diameter was significantly reduced after 18 and 24 months of treatment compared to the value determined before the start of the treatment. 

Tumor diameter in the adenoma group of dogs was subject to statistically significantly greater percentage reductions after 2, 3, 6, 18, and 24 months of antihormonal treatment compared to the carcinoma group (*p* < 0.01). In the epithelioma group of dogs, the percentage reduction in tumor size relative to baseline size reached statistically significant values after 2, 3, and 18 months (*p* < 0.05), and after 6 and 24 months (*p* < 0.01) compared to the carcinoma group of dogs. A statistically significantly greater percentage reduction in tumor diameter occurred in dogs in the adenoma group (no tumor present) after 12 and 18 months of therapy compared to the epithelioma group (*p* < 0.05; Figure 1).

## 4. Discussion

Antihormonal preparations have numerous applications in the treatment of human cancer lesions. Studies show that, if receptors for hormones are present in tumor tissue, antihormonal therapy is justified [17,18]. In cases of breast cancer in women, assessment of the presence of estrogen and progesterone receptors in tumor tissue is a standard procedure in the diagnostic process. Experts recommend immunohistochemical testing for ER and PR to determine whether a lesion is amenable to antihormonal treatment [19]. In our previously published study, we evaluated AR and ER expression in perianal gland tumors in male dogs as potential predictors for treatment with antihormonal therapy with formulations containing tamoxifen (tamoxifen) and cyproterone acetate (Androcur) [11]. Expression of androgen and estrogen receptors was demonstrated in all histological types of the perianal tumors studied, with expression levels for both androgen and estrogen receptor being significantly higher in the adenoma group compared to the epithelioma and carcinoma groups. The study showed that the effect of antihormonal treatment depended on the histological type of the tumor and, thus, the expression levels of AR and ER receptors. Tumors in the adenoma and epithelioma groups, which showed high AR and ER expression in the previous 2021 study, were subject to a significantly greater and faster reduction in diameter over the course of treatment when receiving antihormonal therapy compared to tumors in the carcinoma group. Malignant tumors that showed low AR and ER expression in our 2021 study were less prone to antihormonal treatment than epithelioma and adenoma cases. This allows suggesting that a higher expression of AR and ER in tumor tissue is correlated with a greater effectiveness of antihormonal therapy. 

Tamoxifen therapy is used in human medicine, particularly for the treatment of mammary gland tumors. The effectiveness of treatment depends on the expression of the gene or protein estrogen receptor alpha (ERα) and/or estrogen receptor beta (ERβ). Tamoxifen therapy is commonly used for mammary gland tumors defined as ER-positive, i.e., showing estrogen receptor expression. Low estrogen receptor expression in tumor-lesioned mammary gland cells results in low efficacy of tamoxifen treatment [20,21,22]. ER-positive tumors can be described as having a better prognosis [23], while ER-negative tumors are described as more aggressive and more likely to give metastases [24]. Analogous relationships have been described for mammary gland tumors in female dogs, where multiple receptors—estrogen receptor (ER), progestin receptor (PR), androgen receptor, and receptors for glucocorticoids and/or mineralocorticoids—may be present in the lesioned tissues. Studies have shown that the survival rate of female dogs with mammary carcinoma was significantly higher in receptor-rich (ER and/or PR) tumors [25,26,27,28]. This provides an opportunity to determine prognosis on the basis of receptor expression in tumor tissue. Similar observations were obtained from our study, where tumors showing higher receptor expression (adenomas and epitheliomas) responded positively to antihormonal treatment. The observations made suggest the possibility of using the determination of AR and ER receptor expression in canine perianal gland tumors as predictive indicators of qualification for antihormonal therapy using pharmacological receptor site blockers. Additional confirmation of the effectiveness of the use of tamoxifen in antihormonal therapy stems from studies on 30 dogs with hepatoid gland epitheliomas and adenomas, which demonstrated a decrease in serum VEGF level and a complete remission of neoplastic lesions 1 month after administering tamoxifen [13]. The VEGF levels in dogs with hepatoid gland adenoma continued to decline with time. In the case of dogs with hepatoid gland epithelioma, after the initial drop 1 month after treatment, a rapid increase of the growth factor level was observed, which was significantly higher in animals suffering a relapse of the neoplastic disease (50% of dogs). Changes in the level of VEGF in tamoxifen therapy are consistent with our observations regarding the behavior of the treated neoplastic lesions. However, selective tamoxifen, which is part of standard therapy in females, is not recommended in female dogs due to its partial agonist potential and associated side-effects [29,30]. Studies conducted on healthy, sterilized, and unsterilized female dogs exposed to tamoxifen showed numerous adverse effects. All of the female dogs studied had vulvar swelling and leakage of purulent discharge from the genital tract after 10 days of tamoxifen use at a dose of 0.8 or 0.5 mg/kg BW. Complications of tamoxifen use include purulent discharge (occurred in two female dogs; they underwent ovariohysterectomy and were excluded from further study), vomiting, diarrhea, and decreased appetite [31]. In male dogs treated with tamoxifen, the authors described transient impaired fertility and impaired selected immune parameters [32,33]. The study by Corrada et al. [34] primarily looked at the effect of tamoxifen on the reproductive system in male dogs; testicular size and consistency, libido, testosterone levels, prostate size, and semen quantity and quality were examined, while fertility was also studied after treatment. Tamoxifen was administered for 28 days at a dose of 2.5 mg p.o. A reduction in prostate volume was noted in all tamoxifen-treated dogs, which returned to pretreatment size 5 weeks after the drug was stopped. During the course of treatment, a decrease in testicles, a change in their consistency to soft, a decrease in libido, a decrease in blood testosterone levels, a decrease in semen volume, a decrease in sperm viability, and an increase in sperm morphological abnormalities were observed. All parameters returned to their original values several weeks after tamoxifen treatment was discontinued. Then, 17–20 weeks after the drug was discontinued, the fertility of three male dogs was checked. Each of the three females who were covered by the dogs involved in the study became pregnant [34]. Antihormonal treatment is also applicable to prostate cancer in men. Androgen deprivation therapy (ADT) is commonly used, which involves lowering the levels of androgens responsible for prostate growth [35]. One of the substances used for this purpose is cyproterone acetate [36].

Cyproterone acetate (Androcur) is an androgen receptor antagonist. Because of its ability to block the action of androgens on prostate cells, it is used in men for prostatic hypertrophy and prostate cancer [37,38]. Cyproterone acetate therapy is an effective treatment for prostate cancer in men. Side-effects of therapy have been observed during administration of the drug, including decreased libido, impotence, increased appetite, and weight gain. Cyproterone acetate shows low cardiovascular toxicity [39]. Under the influence of the implemented treatment, the values of PSA (prostate-specific antigen) concentrations steadily decreased within 3–6 months in the range of 30–80%. Furthermore, a decrease in blood testosterone concentrations to post-castration concentrations, as well as an increase in prolactin concentrations, was observed. The volume of the prostate after 1 year of treatment decreased from 30% to 70% [40]. Because of its antagonistic effect against androgen receptors, cyproterone acetate may also be effective in the treatment of other cancers expressing androgen receptors. In our study, antihormonal therapy was applied to dogs diagnosed with neoplastic lesions of the perianal glands of various malignancies. Treatment was carried out with tamoxifen at a dose of 1 mg/kg and Androcur at a dose of 5 mg/kg daily. In the case of benign tumors (adenomas), 90% of the treated lesions went into complete remission, and no recurrence was observed during the 6 month follow-up period. In about 10% of cases, the tumors did not disappear completely but shrank significantly and did not bleed; hence, in such cases, treatment was extended for another 2–3 weeks, after which the benign tumors disappeared completely. Over a period of 6 months of further observation, there was no recurrence of the neoplastic process. Epitheliomas of low malignancy completely disappeared in 70% of cases after 1 month of therapy, whereas, in 30% of cases, they only decreased in diameter, and prolonging therapy did not lead to further shrinkage of the tumors. At 6 month follow-up, 60% of epitheliomas recurred. Carcinomas decreased in diameter during treatment, and the breakdown of tumor tissue was inhibited, but the tumors did not disappear completely. Similar results were obtained by Tozon et al. [10,15] studying the effectiveness of treating tumors of the perianal region with castration and antihormonal preparations. They showed that this method is not effective for malignant tumors. These results are in line with the results of our study, where malignant tumors also failed to respond to antihormonal treatment, as, despite the reduction in tumor size, they did not completely disappear. 

Studies with humans show that estrogen receptor expression is even lower in the nuclei of highly malignant tumors than in tumor-free tissues. Consequently, treatment of such tumors with estrogen receptor antagonists has not been successful, and metastasis has often occurred. In contrast, treatment of tumors, with high expression of estrogen receptors alpha and beta, with agents that act as antagonists to estrogen receptors (tamoxifen, raloxifene) had a satisfactory effect [41]. Our research, in which the expression of AR and ER receptors in tumor tissue was determined in the first stage taking into account the number of cells having receptors and the intensity of their staining, showed that, in benign and low-malignancy tumors (adenoma and epithelioma), more cells show expression of AR and ER receptors than in malignant tumors (carcinoma). Similarly, in cases of adenoma and epithelioma, the intensity of receptor staining was higher than in carcinoma. These results are reflected in the obtained effects of the applied antihormonal therapy, which proved to be more effective in cases of benign and low-malignancy tumors, as opposed to malignant tumors not amenable to antihormonal treatment. On the other hand, the positive effect of antihormonal treatment of malignant tumors was a reduction in their size and a reduction in the processes of tumor tissue breakdown and bleeding of the affected tissues, which are among the most troublesome symptoms of the disease for the owner and the animal. 

## 5. Conclusions

In summary, the results show that hepatoid adenomas characterized by high expression of AR and ER receptors respond positively to antihormonal therapy with drugs that block receptor sites, and treatment leads to complete tumor regression. For locally malignant tumors of the epithelioma type and carcinomas with low expression of AR and ER receptors, antihormonal therapy makes it possible to reduce the size of the tumor, but does not make it possible to cure it completely.

## Figures and Tables

**Figure 1 animals-13-00463-f001:**
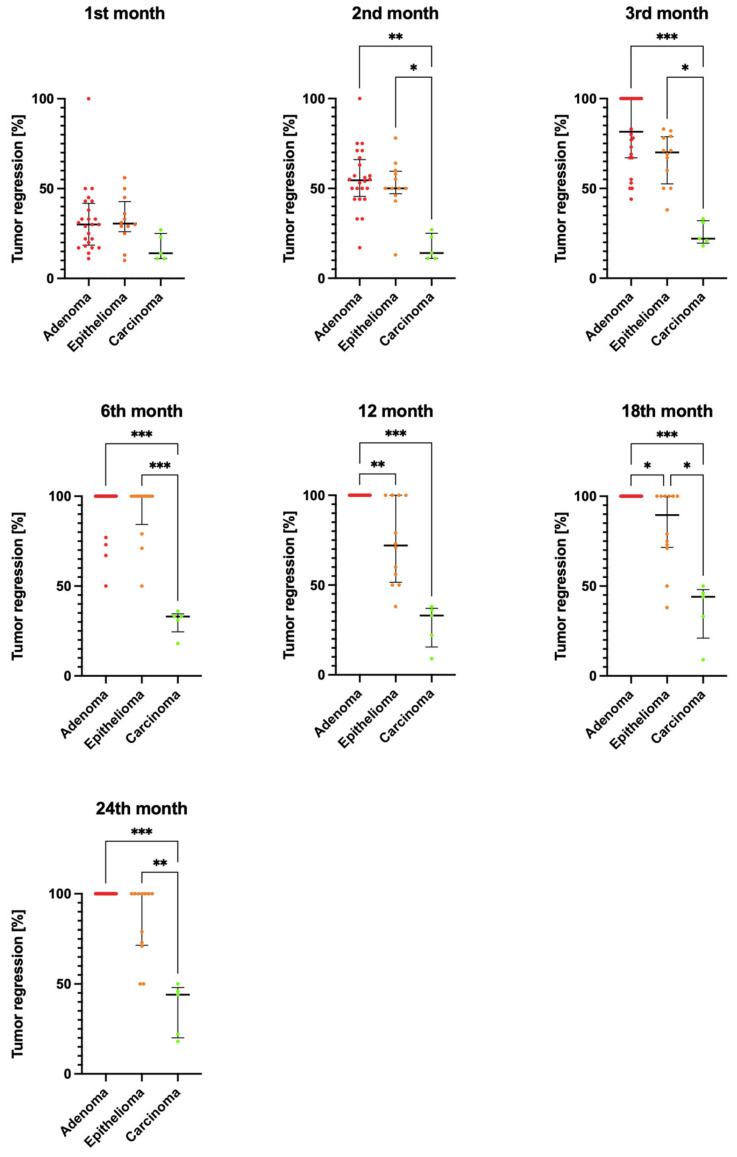
Comparison of the percentage reduction in tumor diameter between the groups: adenoma (*n* = 24), epithelioma (*n* = 12), and carcinoma (*n* = 5) after 1, 2, 3, 6, 12, 18, and 24 months of antihormonal treatment. Data were analyzed using the Kruskal–Wallis test. The median is shown as a horizontal bar. The whiskers span the interquartile range. Significant differences were determined by comparing tumor sizes between experimental groups within each treatment interval; * *p* < 0.05, ** *p* < 0.01, and *** *p* < 0.001.

**Table 1 animals-13-00463-t001:** Tumor diameter in dogs, measured in millimeters, before treatment (baseline) and after 1, 2, 3, 6, 12, 18, and 24 months of antihormonal treatment in the adenoma (*n* = 24), epithelioma (*n* = 12), and carcinoma (N = 5) experimental groups.

	Adenoma	Epithelioma	Carcinoma
Before treatment (baseline)	9.00 (4.00)	10.00 (4.75)	11.00 (4.50)
After 1 month of treatment	6.00 (4.50)	6.50 (3.75)	8.00 (3.00)
After 2 months of treatment	5.00 (3.50)	5.00 (2.50)	8.00 (3.00)
After 3 months of treatment	2.00 (3.00) ***	3.50 (2.00) **	9.00 (3.50)
After 6 months of treatment	0.00 (0.00) ***	0.00 (2.25) ***	9.00 (3.00)
After 12 months of treatment	0.00 (0.00) ***	3.50 (4.00) ***	8.00 (3.00)
After 18 months of treatment	0.00 (0.00) ***	1.50 (3.75) ***	7.00 (3.00) *
After 24 months of treatment	0.00 (0.00) ***	0.00 (3.75) ***	7.00 (2.00) *

Data were analyzed using the Friedman test for multiple comparisons. Data are expressed as the median with the interquartile range (presented in brackets). Significant differences were determined by comparing tumor sizes during/after treatment to the baseline tumor size within each experimental group; * *p* < 0.05, ** *p* < 0.01, and *** *p* < 0.001.

## Data Availability

The datasets used and analyzed in the current study are available from the corresponding authors on reasonable request.

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
