# Peer review of "Pharmacological Treatment of Perianal Gland Tumors in Male Dogs"

_animals, 2023, doi:10.3390/ani13030463_

Round 1
Reviewer 1 Report
The paper deals with an important clinical problem of neoplastic tumours of the perineal region. It is all the more interesting because it seeks conservative treatment options instead of surgical therapy.
[67] please write what drugs were used
[88] please write what breed of dogs were treated
[88] Were the dogs castrated as part of their treatment and only conservative therapy used?
[104] please change to 1 mg/kg BW
[105] please change to 5 mg/kg BW
[106] do you use electronic callipers?
[106] who measured the tumour? Was it always the same person?
[186]-[201] please support these statements with relevant publication
Author Response
Thank you very much for taking the time to review our article. Below you will find answers to comments and suggestions.
[67] please write what drugs were used
In accordance with the reviewer's comments, the name of the drug and publications have been added.
The described tumor lesions were tried to be treated with the use of antihormonal preparations such as tamoxifen citras [Brodzki 2019, Sobczyńska-Rak 2014].
[88] please write what breed of dogs were treated
As noted by the reviewer, in the “Material and methods” chapter, the number and percentage content of dogs of individual breeds has been introduced.
The study was performed on 41 male, non-orchidectomized dogs aged 4-17 years. Mixed breed dogs - 16 (67%), Cocker Spaniels – 6 (15%), Wire-Haired Fox Terriers – 3 (7%), Dachshunds – 2 (5 %), German shepherd - 2 (5%) and Miniature Pinscher – 2 (5%) were the most common breeds in this study. The remaining breeds were represented by one individual.
[88] Were the dogs castrated as part of their treatment and only conservative therapy used?
Answering the reviewer's question, I inform that only non-orchidectomized dogs were included in the study. The treated dogs were not castrated but pharmacologically treated only. An appropriate correction was introduced in the first sentence of the “Material and methods” chapter
The study was performed on 41 male, non-orchidectomized dogs aged 4-17 years.
[104] please change to 1 mg/kg BW
[105] please change to 5 mg/kg BW
We agree with this suggestion. The sentence was corrected as follows:
Treatment with the active substance tamoxifen (Tamoxifen preparation), at a dose of 1 mg/kg BW orally in tablet form, and cyproterone acetate (Androcur) at a dose of 5 mg/kg BW also orally, was used.
[106] do you use electronic callipers?
Yes, we used electronoc calipers.
[106] who measured the tumour? Was it always the same person?
The measurements were always made by the same person. The sentence was corrected as follows:
During the course of the study, tumor diameters were measured regularly with electronic calipers by te same person and recorded in millimeters starting with the measurement before treatment, then after 1, 2, 3, 6, 12, 18 and 24 months of therapy.
[186]-[201] please support these statements with relevant publication
As the reviewer pointed out, the sentence was supplemented by the relevant publication.
In our previously published study from we evaluated AR and ER expression in perianal gland tumors in male dogs as potential predictors for treatment with antihormonal therapy with formulations containing tamoxifen (tamoxifen) and cyproterone acetate (androcur) [Brodzki 2021].
Thank you very much for your valuable comments and suggestions that contributed to increasing the value of the article.
Reviewer 2 Report
In principle an interesting study is presented. The question if the different types of perianal gland tumours will respond to anti-hormonal treatment is an important question. However, in this study a lot of information is missing which is important to fully interpret the results of the therapy. No information on the castration status, and its relationship to the different subtypes. No information on the hormone receptor status (AR/PR) of the tumours, and finally the reliability of histological classification into benign and malignant is not optimal based on core biopsies. This latter should have been addressed in the Discussion.
Line 36-37: “The results show that hepatoid adenomas that are characterized by high expression of AR and ER positively respond to antihormonal therapy” In this study AR and ER were not investigated and therefore this statement cannot be made. The same for the carcinomas in the next sentence.
Line 101: It is questionable if it will be possible to make a reliable histological malignancy grading based on a histological biopsy as one of the most important malignancy criteria in perianal gland tumours is the invasiveness of tumours cells into adjacent tissues (Stannard & Pulley 1978).
Line 114: different experimental groups were compared, but it has not been explained what these different experimental groups are.
Line 119: three different levels of significance are being presented. This is confusing. Is there a correction for multiple testing performed (e.g. Bonferroni)? If not, than probably only P<0.001 will be a real significant level considering the multiple testing performed.
Line 121 Results: as these tumours are hormone dependent and are being treated with hormones, it is of the utmost importance to know the number of male/female dogs, castration status and the relation with the histological subtype. In addition, the ER/AR status of the tumours of these dogs is important.
Table 1 and 2: suddenly a histological classification into adenomas, carcinomas and epitheliomas is presented, without introduction of these different histological types. In addition, it makes more sense to change the order and put the epitheliomas between the adenomas and the carcinomas, according to their biological behavior. Please also explain the numbers that are presented in brackets in the legend of the Table.
Line 136: Table 2 does not add much. It has the same information as in Table 1, but know expressed as percentage reduction. It can be deleted and substituted in the text by mentioning when maximum reduction was seen. Figure 1 also presents the same data as Table 1 but now visual. In addition, all information of Table 1 and Table 2 is written down in the text.
Fig 2: This gives the same information as Fig 1, but than on an individual base. It can be deleted or given as an Addenda figure.
Line 186: apart from the year 2021 please give reference.
Discussion: Far too much emphasis is given to the hormone expression and into-hormonal treatment of human mammary tumours. This could be much shorter. Also the rest of the discussion could be more concise. In the discussion on the the anti-tumor effect of tamoxifen on perianal gland tumors, however, important information is missing (“A recent study in 30 dogs with hepatoid gland epitheliomas and adenomas, however, demonstrated a decrease in serum VEGF level and a complete remission of neoplastic lesions one month after administering tamoxifen (Sobczyńska-Rak & Brodzki 2014). The VEGF levels in dogs with hepatoid gland adenoma continued to decline with time. In the case of dogs with hepatoid gland epithelioma, after the initial drop one month after treatment, a rapid increase of the growth factor level was observed, which was significantly higher in animals suffering a relapse of the neoplastic disease (50% of dogs). “)
Conclusions: see my earlier remark on the fact that conclusions based on relation to AR/ER status cannot be made due to absence of hormone status of the tumours in the present study and the missing of castration status of the animals.
Author Response
Thank you very much for taking the time to review our article. Below you will find answers to comments and suggestions.
Line 36-37: “The results show that hepatoid adenomas that are characterized by high expression of AR and ER positively respond to antihormonal therapy” In this study AR and ER were not investigated and therefore this statement cannot be made. The same for the carcinomas in the next sentence.
Thank you very much for the important remark.
The previous version of the manuscript does not make it clear that authors have already published a work on the expression of AR and ER receptors in canine perianal tumors. The same dogs were included in the current study. The "Material and methods" section has been redrafted to make it clear that the tumors from tested dogs had previously undergone histopathology and immunohistochemistry and AR and ER receptor expression had been evaluated.
This part of the description is currently as follows:
The animals were patients treated for neoplastic lesions of the anal region at the Department of Animal Surgery of the University of Life Sciences in Lublin. After taking a history and performing a clinical evaluation of the lesions, tumor sections were taken from the diagnosed animals for histopathological examination and immunohistochemical assessment of AR and ER receptors using a 0.6 cm diameter skin trepan. Histopathological examination carried out based on the histological classification of skin tumors according to AFIP/WHO revealed 24 adenomas, 12 epitheliomas and 5 carcinomas [Goldschmidt 1998]. The immunohistochemical staining showed mainly nuclear expression of AR and ER in the neoplastic cells. Both the androgen and estrogen receptors were expressed in adenoma, epithelioma and carcinoma cases. However, the highest expression of the receptors was stated in adenoma and then in epithelioma representing a low-grade malignant neoplasms. In case of carcinoma, the expression of sex hormone receptors was very weak. Detailed results of histopathological examination and immunohistochemical determinations of AR and ER receptors in tumors from dogs included in the present study have been published [Brodzki 2021].
Line 101: It is questionable if it will be possible to make a reliable histological malignancy grading based on a histological biopsy as one of the most important malignancy criteria in perianal gland tumours is the invasiveness of tumours cells into adjacent tissues (Stannard & Pulley 1978).
Thank you very much for this critical remark. However, I would like to point out that in the assessment of malignancy, we take also into account other features such as: lack of organization in lobules, nuclear pleomorphism, presence of mitoses in basaloid and hepatoid cells as well mitotic count, reactive collagenous stroma and lymphatic invasion. In our case, the histopathological diagnosis of hepatoid carcinoma was consistent with the estimation of AR and ER expression (very low expression) and the clinical course of the disease.
Line 114: different experimental groups were compared, but it has not been explained what these different experimental groups are.
Thank you very much for this critical remark. The “Material and methods” section has been supplemented with the following sentence:
After histopathological evaluation of the lesions, antihormonal, pharmacological therapy was implemented. The animals were divided into three groups depending on the results of histopathological examination, i.e. adenoma, epithelioma and carcinoma group.
Line 119: three different levels of significance are being presented. This is confusing. Is there a correction for multiple testing performed (e.g. Bonferroni)? If not, than probably only P<0.001 will be a real significant level considering the multiple testing performed.
Data were analyzed using the Friedman test for multiple comparisons with the corrected Dunn’s test (Bonferroni correction) as a post-hoc test. We presented three different levels of significance (p< 0.05 and p< 0.01 and p< 0.001) which is a common way of p-value presentation in biomedical papers. We can’t be strict only to p< 0.001 because it would be required rejection of the significance of the results with p< 0.05 and p< 0.01. However, we can present only p< 0.05 without specifying which significant results have lower p-value levels than 0.01 and 0.001.
Line 121 Results: as these tumours are hormone dependent and are being treated with hormones, it is of the utmost importance to know the number of male/female dogs, castration status and the relation with the histological subtype. In addition, the ER/AR status of the tumours of these dogs is important.
Thank you for this reviewer's suggestion.
As mentioned earlier, the hormonal status of AR and ER receptors in particular groups of tumors was previously determined by immunohistochemistry and published [Brodzki et al. 2021]. Only non-orchidectomized, male dogs were included in the study. No castration was used in the treatment, only pharmacological treatment. The “Material and methods”section has been redrafted and appropriate information has been included.
Table 1 and 2: suddenly a histological classification into adenomas, carcinomas and epitheliomas is presented, without introduction of these different histological types. In addition, it makes more sense to change the order and put the epitheliomas between the adenomas and the carcinomas, according to their biological behavior. Please also explain the numbers that are presented in brackets in the legend of the Table.
Thank you for this reviewer's suggestion.
The classification of perianal tumors was based on Goldschmidt M.H. et al “Histological classification of epithelial and melanocytic tumors of the skin of domestic animals”. WHO/AFIP 1998. This classification contains detailed descriptions of individual histological types. Therefore, the description of individual tumors was omitted. We fully agree with the reviewer's suggestion to change the order of individual tumors in the tables and figures from adenoma to epithelioma and carcinoma. The order has been changed in the revised tables and figures. Data in tabels are expresssed as median with an interquartile range (presented in brackets).
Line 136: Table 2 does not add much. It has the same information as in Table 1, but know expressed as percentage reduction. It can be deleted and substituted in the text by mentioning when maximum reduction was seen. Figure 1 also presents the same data as Table 1 but now visual. In addition, all information of Table 1 and Table 2 is written down in the text.
We suggest, however, to leave both tables because they present the same data but analyzed differently. See answer for Fig 2. please.
Fig 2: This gives the same information as Fig 1, but than on an individual base. It can be deleted or given as an Addenda figure.
In our opinion none of the figures can be removed. Figures show the same date, however, analyzed differently. Figure 1. shows the comparison of tumor sizes during/after treatment to the baseline tumor size within each experimental group (Friedman test for multiple comparisons). Fgure 2. shows the comparison of tumor sizes between experimental groups within each treatment interval (Kruskal-Wallis test).
Line 186: apart from the year 2021 please give reference.
Reference added as noted.
Discussion: Far too much emphasis is given to the hormone expression and into-hormonal treatment of human mammary tumours. This could be much shorter. Also the rest of the discussion could be more concise. In the discussion on the the anti-tumor effect of tamoxifen on perianal gland tumors, however, important information is missing (“A recent study in 30 dogs with hepatoid gland epitheliomas and adenomas, however, demonstrated a decrease in serum VEGF level and a complete remission of neoplastic lesions one month after administering tamoxifen (Sobczyńska-Rak & Brodzki 2014). The VEGF levels in dogs with hepatoid gland adenoma continued to decline with time. In the case of dogs with hepatoid gland epithelioma, after the initial drop one month after treatment, a rapid increase of the growth factor level was observed, which was significantly higher in animals suffering a relapse of the neoplastic disease (50% of dogs). “)
Thank you very much for your valuable suggestion. The discussion was supplemented with conclusions from the article by Sobczyńska-Rak and Brodzki 2014
Conclusions: see my earlier remark on the fact that conclusions based on relation to AR/ER status cannot be made due to absence of hormone status of the tumours in the present study and the missing of castration status of the animals.
In our opinion, the additional information introduced in accordance with the reviewer's comments justifies the presented conclusions.
Thank you very much for your valuable comments and suggestions that contributed to increasing the value of the article.
Round 2
Reviewer 1 Report
The paper has been revised. The authors have taken into account the suggestions and added the correct information to the text.In the current version, the paper can be published.
Author Response
Dear Reviewer,
thank you very much, once again, for the corrections and suggestions that contributed to increasing the value of our article.
Reviewer 2 Report
The manuscript has improved by the clarifications of the authors and the addition of the hormone receptor work. There are, however, still two items in which is disagree with the authors.
I still have problems with the two tables and two figures. Table 2 is giving the same information one can get from Table 1 and should be deleted. Figure 2 is also giving the same information as figure 1 and should be deleted.
The discussion is quite long and the part on hormone receptors in mammary tumours should be deleted as it had nothing to do with the study and is a completely different tumour from the perianal gland tumours.
Author Response
Dear Reviewer,
thank you once again for the corrections and suggestions that contribut to increasing the value of our article.
After reconsidering the suggestion to reduce the number of tables and figures, we propose to keep Table 1 and Figure 2 and remove Table 2 and Figure 1 from the manuscript. An appropriate corrections have been made in the manuscript.
The authors believe that the removal of the excerpt about breast cancer would have a negative impact on the quality of the entire work. Due to the small number of publications on the antihormonal treatment of tumors in domestic animals, the model of therapy in humans was used in the discussion. The division of neoplastic lesions into "positive" and "negative" depending on the expression of hormone receptors in breast tumors in women was considered by the authors to be transferable to cases of the perianal gland tumors and used in the diagnostic process in order to select the appropriate therapy. In cases of breast cancers, similarly to tumors of the perianal glands, in the case of intensive expression of hormone receptors, higher effectiveness of antihormonal treatment was confirmed comparing to tumors with low expression of hormone receptors. In addition, the breast cancer treatment protocols use the same preparation that the authors used in their studies to treat the dogs. For these reasons, the authors see significant similarities between the two types of tumors, despite the obvious differences between them. We therefore propose to keep this part unchanged in the discussion.